# Primary Processes of Free Radical Formation in Pharmaceutical Formulations of Therapeutic Proteins

**DOI:** 10.3390/biom13071142

**Published:** 2023-07-17

**Authors:** Christian Schöneich

**Affiliations:** Department of Pharmaceutical Chemistry, University of Kansas, 2093 Constant Avenue, Lawrence, KS 66047, USA; schoneic@ku.edu; Tel.: +1-(785)-864-4880

**Keywords:** autoxidation, Fenton reaction, free radicals, mechanisms, oxidation, photo-degradation, cavitation

## Abstract

Oxidation represents a major pathway for the chemical degradation of pharmaceutical formulations. Few specific details are available on the mechanisms that trigger oxidation reactions in these formulations, specifically with respect to the formation of free radicals. Hence, these mechanisms must be formulated based on information on impurities and stress factors resulting from manufacturing, transportation and storage. In more detail, this article focusses on autoxidation, metal-catalyzed oxidation, photo-degradation and radicals generated from cavitation as a result of mechanical stress. Emphasis is placed on probable rather than theoretically possible pathways.

## 1. Introduction

The advent of biotechnology has enabled the production of recombinant proteins for therapeutic applications. A recent review of the globally highest-selling drugs in 2019 showed that out of ten drug products, seven were proteins [1]. Despite the therapeutic and commercial success of protein therapeutics, the development of stable protein formulations can present challenges [2,3,4,5,6,7]. Proteins are subject to physical and chemical degradation, potentially compromising the efficacy and safety of drug products. The physical degradation of proteins is often associated with processes such as surface adsorption, aggregation, particle formation and precipitation, while chemical degradation describes the covalent modification of amino acids. Frequently, the physical and chemical degradation of proteins are connected, where, for example, chemical modifications may trigger aggregation or conformational transitions of proteins may facilitate the accrual of chemical modifications.

Oxidation represents a major pathway for the chemical degradation of proteins, which can be carried out by a range of reactive oxygen and nitrogen species, including free radicals [8,9]. The field of redox biology presents many examples of proteins that are subject to oxidative modification in vivo under conditions of oxidative stress. These oxidative modifications may either result in no change in activity, or promote loss or gain of function, depending on the nature of the modifications and the specific proteins. Whether some of these oxidative modifications may be useful as clinical biomarkers will depend on the type, stability and location of the modifications and the pathologies of concern [10,11,12]. For example, commonly measured protein oxidation products such as protein carbonyls, methionine sulfoxide (MetSO) and some tyrosine-derivatives were poor biomarkers in the biological fluids of rats for either carbon tetrachloride (CCl_4_)- or ozone-induced oxidative stress [13,14]. In contrast, some lipid-derived oxidation products, such as malondialdehyde (MDA) or isoprostanes, appeared to be viable biomarkers for CCl_4_-induced oxidative stress in rats [13].

Many of the protein oxidation products that have been characterized in vivo can also form as a result of oxidative processes in therapeutic protein formulations in vitro [15]. In addition, these oxidation processes can generate a range of oxidation products from excipients, e.g., from amino acids, especially histidine (His) [16,17,18,19], and surfactants [18,19,20,21,22,23,24], and likely also carbohydrates. Some of these oxidation products in vitro may correlate with important characteristics of their respective drug products, and are referred to as critical quality attributes (CQAs). For example, oxidation products may have consequences for the shelf-life, bioavailability or immunogenicity of drug products.

The exact nature and sources of oxidants in protein formulations are generally less well defined. For comparison, the biological mechanisms of oxidant production frequently rely on relatively well-characterized enzymes such as xanthine oxidase [25], nitric oxide synthase [26], myeloperoxidase [27] or NADPH oxidase [28]. In contrast, oxidation reactions in therapeutic protein formulations in vitro rely predominantly on adventitious processes promoted by, e.g., mechanical stress, impurities and/or exposure to elevated temperature, light or ionizing radiation. Details on the primary processes that lead to free radical formation and oxidation in pharmaceutical formulations would be highly valuable for the development of mitigation strategies. It is possible to outline the mechanisms of free radical generation in pharmaceutical formulations based on information on impurities and known stress factors that are relevant to pharmaceutical manufacturing, transportation and storage. This is the purpose of this article.

## 2. Composition of Pharmaceutical Formulations of Therapeutic Proteins

The pharmaceutical formulations of therapeutic proteins display a range of compositions for liquid, frozen and lyophilized forms. Relevant to the potential formation mechanisms of radicals is the fact that these formulations can generally contain several classes of compounds in addition to the protein, such as buffers, surfactants, amino acids, cryoprotectants, chelators and additional tonicifiers [1,29]. Besides the intended functions in the formulations, each of these respective components may play a role in free radical generation through its chemical properties and/or impurities, which may be introduced via chemical synthesis, purification and/or storage.

## 3. Pathways of Free Radical Formation That Are Relevant to Pharmaceutical Formulations

The following sections will discuss specific pathways of free radical generation with respect to the potential role of impurities and stress factors.

### 3.1. Autoxidation

Miller et al. define true autoxidation “as the spontaneous oxidation in air of a substance not requiring catalysts” [30]. Hence, autoxidation can be represented by the general reaction (1), where D^−^ and O_2_^•−^ represent an electron donor and superoxide, respectively, and k_1_ and k_−1_ are the rate constants for forward and reverse electron transfer.
D^−^ + O_2_ ⇌ D^•^ + O_2_^•−^(1)

A plot of log k_1_ vs. log K_1_ (where K_1_ represents the equilibrium constant for redox equilibrium 1) yields a curve that can be fitted to the Marcus equation, where D^−^ represents a series of phenolates, indophenolates and other electron donors [31]. This relationship allows us to make an estimate of the sensitivity of amino acid side chains towards autoxidation on the basis of their reduction potentials. Such an estimate suggests that at most cysteine in its deprotonated form, with E^o^_2_ ≈ 0.75 V for redox equilibrium 2 [32,33], would be susceptible to autoxidation at pH values generally selected for protein formulations.
RS^•^ + e^−^ ⇌ RS^−^(2)

This would limit autoxidation processes to proteins that contain free cysteine residues. Monoclonal antibodies do not contain free cysteine residues, except for small quantities of incompletely folded proteins, implying that autoxidation should be a negligible problem for pharmaceutical formulations containing monoclonal antibodies.

A second target for potential autoxidation would be surfactants [34], especially polysorbate 80, which contains oleic, linoleic and linolenic acid [35,36]. It is possible that autoxidation contributes to the generation of polysorbate radicals [37] and polysorbate oxidation products, including peroxides, in neat polysorbate [38]. However, it is equally likely that oxidation in neat polysorbate is triggered by the homolytic decomposition of reactive fatty acid:oxygen copolymers of the general structure **1** (Figure 1, where residues R_n_, n = 1–6, depict moieties of fatty acids that have undergone successive peroxyl radical and oxygen addition to double bonds) [39,40], containing α,β-diperoxide repeats analogous to styrene:oxygen copolymers [41]. Such fatty acid:oxygen copolymers may be generated during polysorbate synthesis and storage.

Morita and Tokita reported that fatty acid:oxygen copolymers are stronger initiators of lipid peroxidation in model experiments compared to simple hydroperoxides [39]. However, the fact that Bensaid et al. [42] observed that iron levels as low as 20 ppb accelerated polysorbate oxidation in aqueous formulations suggests that, at least in pharmaceutical formulations, metal-catalyzed reactions of hydroperoxides may be kinetically more significant for the formation of free radicals compared to metal-independent decomposition reactions of fatty acid:oxygen copolymers or autoxidation. This is also consistent with data showing the accelerated decomposition of polysorbate when in contact with stainless steel surfaces [19,43]. It is, therefore, unlikely that true autoxidation processes contribute significantly to free radical formation in pharmaceutical formulations.

### 3.2. Fenton and Fenton-like Reactions between Metals and Peroxides

Fenton and Fenton-like reactions represent important pathways for free radical generation. Peroxides can be introduced into formulations through excipients [44], primarily surfactants [38,44,45], and/or as a result of sterilization procedures [46]. Before presenting a detailed discussion of the potential radical-generating reactions of metals and peroxides in pharmaceutical formulations, we need to evaluate which metals and which reactions are most relevant to pharmaceutical formulations. The International Council for Harmonization (ICH) Q3D(R2) guidelines define three classes of elemental impurities based on “their toxicity (PDE) and likelihood of occurrence in the drug product” [47] (PDE = permitted daily exposure). Several elemental impurities in these classes are redox-active and/or catalyze oxidation reactions, such as Co, Ni and V (in class 2A); Ir, Os, Rh and Ru (in class 2B); and Cr, Cu and Mo (in class 3). The ICH Q3D(R2) guidelines list additional elemental impurities “for which PDE values have not been established due to their low inherent toxicity and/or differences in regional regulations” [47]. Of these, Fe, Mn and W are redox-active and/or catalyze oxidation reactions (as, perhaps, does Al [48]; here, Al^III^ does not change its oxidation state but promotes the disproportionation reaction of two complexed H_2_O_2_ molecules). A representative quantitative analysis of the elemental impurities listed in class 1 and 2A in several formulation components predicts that their levels in therapeutic protein drug products will be significantly below the PDE [49]. These elemental impurities will likely present no toxicological problems; however, even at levels below the PDE, some of these elemental impurities may promote radical formation and/or catalyze oxidation reactions. Therefore, we need to narrow down a selection of metals for further consideration in this article via other means. Class 2B elements “have a reduced probability of occurence in drug product” [47] and will, therefore, not be considered further. Lloyd et al. reported DNA oxidation in the presence of H_2_O_2_ for Cr(III), Fe(II), V(III) and Cu(II), indicating Fenton or Fenton-like reactivities of these metals [50]. However, no efficient DNA oxidation was observed for Co(II) and Ni(II) in the presence of H_2_O_2_ [50]. Anipsitakis and Dionysiou surveyed the formation of radicals from the reaction of three oxidants, including hydrogen peroxide (H_2_O_2_), potassium persulfate (K_2_S_2_O_8_) and potassium peroxomonosulfate (KHSO_5_), with nine metals, including Fe(II), Fe(III), Co(II), Ru(III), Ag(I), Ce(III), V(III), Mn(II) and Ni(II) [51]. Of these, only Fe(II), Fe(III) and Ru(III) generated significant levels of hydroxyl radicals (HO^•^) upon reaction with H_2_O_2_ [51]. The other metals formed significant yields of inorganic radicals (SO_4_^•−^) only upon reaction with K_2_S_2_O_8_ and/or KHSO_5_, oxidants that are likely not present in pharmaceutical formulations. In contrast, Stadtman et al. advocate for the formation of “caged” HO^•^ radicals during the reaction of Mn(II) with H_2_O_2_, available for the oxidation of substrates [52,53,54]. The data are consistent with respect to Co(II) and Ni(II), which we do not need to consider further as a source of radicals in pharmaceutical formulations. We will also not further consider any reactions of Mo(IV) as it functions as a co-catalyst in Fe-dependent Fenton reactions [55,56], for example, reducing Fe(III) to Fe(II) [55]. The formation and detection of HO^•^ radicals during reactions of V(III) and Mn(II) with H_2_O_2_ may depend on the experimental conditions; in a first approximation, the redox reactions of V(III) and Mn(II) with H_2_O_2_ would be rather comparable to the reactions of the Fe(II)/Fe(III) redox couple with H_2_O_2_. Hence, a more detailed description of the reaction mechanisms of Fe(II) and Fe(III) would serve as a model for analogous reactions of V(III) and Mn(II), and also of Cr(III) and Cu(II). Therefore, the following will entirely focus on processes of Fe-dependent radical generation through the Fenton reaction that are relevant to pharmaceutical formulations.

#### Reactions of Ferrous and Ferric Iron

In pharmaceutical formulations of therapeutic proteins, iron impurities can come from multiple sources, including the cell culture medium, manufacturing equipment, containers, proteins and excipients. Iron levels as high as 1–9 μM have been reported for some protein formulations [57,58].

In general, it can be assumed that iron impurities in pharmaceutical formulations will be present as ferric iron, Fe^III^, coordinated with iron-binding ligands, L [59]. These ligands can originate from the protein as well as excipients such as amino acids and carbohydrates. Based on the formulation composition, it is likely that Fe^III^ may be present in a variety of mixed ligand complexes, i.e., that complexes of Fe^III^ show some heterogeneity. Fe^III^ reacts with H_2_O_2_ according to equilibrium 3 [60,61].
L_x_Fe^III^ + H_2_O_2_ ⇌ L_x_Fe^III^(^−^O_2_H) + H^+^(3)

Rate constants of k_3_ = 69 M^−1^s^−1^ and k_−3_ = 0.11 s^−1^ have been reported for equilibrium 3 in acidic aqueous solution with pH 2.0 (where L = H_2_O) [61]. Based on the standard reduction potentials for the couples Fe^III^/Fe^II^ (0.77 V vs. NHE) [62] and HO_2_/HO_2_^−^ (0.79 V), the reduction of Fe^III^ by HO_2_^−^ is feasible [63], so equilibrium 4 is reasonable, where the resulting hydroperoxyl radical is characterized by pK_a_ = 4.8 for equilibrium 5 [64]. The superoxide radical anion (O_2_^•−^) can subsequently reduce an additional equivalent of L_x_Fe^III^ (reaction 6) [62,65].
L_x_Fe^III^(^−^O_2_H) ⇌ L_x_Fe^II^ + HOO^•^(4)
HOO^•^ ⇌ H^+^ + O_2_^•−^(5)
L_x_Fe^III^ + O_2_^•−^ ⇌ L_x_Fe^II^ + O_2_(6)

However, it has been pointed out that, specifically, the reduction potential for the couple Fe^III^/Fe^II^ is very sensitive to pH [59] and the nature and concentration of the ligands [62,63], so the potential reduction of L_x_Fe^III^ to L_x_Fe^II^ by HO_2_^−^ must be carefully discussed with respect to these parameters. Specifically, for L_x_ = EDTA, the reduction potential of Fe^III^/Fe^II^ decreases to 0.12 V [62], suggesting that the reduction of (EDTA)Fe^III^ by HO_2_^−^ may not be a major pathway of Fe^II^ formation [63] (for L_x_ = DTPA, the reduction potential decreases even further to 0.03 V [66]). However, this prediction must be compared to experimental results that show that the reaction of H_2_O_2_ with (EDTA)Fe^III^ yields an oxidant that converts the dipeptide Met-Met (Met = methionine) to products that are also generated via the exposure of Met-Met to a Fenton system, (EDTA)Fe^II^/H_2_O_2_ [67], suggesting the formation of free or complexed hydroxyl radicals (HO^•^) or higher-valent iron-oxo species such as Fe^IV^=O [61]. In this respect, the results of Bensaid et al. [42] are important, which show that the levels of iron impurities (20 ppb vs. <2 ppb) in formulations containing a monoclonal antibody, His, sucrose and polysorbate 80 control polysorbate oxidation, which correlates with the oxidation of Met^255^ on the monoclonal antibody. In these formulations, it is likely that HO^•^ and/or L_x_Fe^IV^=O are generated via the reaction of L_x_Fe^III^ with hydrogen peroxide (reactions 3, 4, 7 and 8) and RO^•^ and/or L_x_Fe^IV^=O via the reaction of L_x_Fe^II^ with organic hydroperoxide impurities (reactions 10 and 11). Here, L_x_Fe^IV^=O (E^o′^_pH 7.0_ ≈ 1.00 V) is the less powerful and more selective oxidant compared to HO^•^ (E^o′^_pH 7.0_ = 2.18 V) [68].
L_x_Fe^II^ + H_2_O_2_ → L_x_Fe^III^ + HO^•^ + HO^−^(7)
L_x_Fe^II^ + H_2_O_2_ → L_x_Fe^IV^=O + H_2_O(8)
L_x_Fe^III^(^−^O_2_R) ⇌ L_x_Fe^II^ + ROO^•^
(9)
L_x_Fe^II^ + HOOR → L_x_Fe^III^ + RO^•^ + HO^−^(10)
L_x_Fe^II^ + HOOR → L_x_Fe^IV^=O + ROH(11)

In this regard, the initial reaction (reaction 3) of L_x_Fe^III^ with H_2_O_2_ may become important, as its product, L_x_Fe^III^(^−^O_2_H), reacts significantly more efficiently with L_x_Fe^II^ (k_12_ = 7.7 × 10^5^ M^−1^s^−1^; L = H_2_O, pH 1.0) compared to H_2_O_2_ (k ≈ 50 M^−1^s^−1^) [61].
L_x_Fe^III^(^−^O_2_H) + L_x_Fe^II^ → L_x_Fe^III^ + [L_x_Fe^III^ + HO^•^]/L_x_Fe^IV^=O(12)

Ultimately, the resulting oxidizing species, HO^•^ and/or L_x_Fe^IV^=O, will have the opportunity to react with formulation constituents such as protein and excipients, generating a plethora of oxidation products and secondary oxidizing species including peroxyl radicals, alkoxyl radicals and peroxides. The initial oxidation reactions of HO^•^ and/or L_x_Fe^IV^=O may occur preferentially with the ligands L, coordinating either Fe^II^ or Fe^III^ [61]. Peroxyl and alkoxyl radicals, as well as peroxides, will be generated via the reaction of HO^•^ and/or L_x_Fe^IV^=O with the organic constituents of the formulation. An alternative oxidant, the carbonate radical anion (^•^CO_3_^−^), may be generated if the formulation contains low amounts of bicarbonate (introduced through atmospheric CO_2_), which can generate L_x_Fe^II^(CO_3_) [69]. In such complexes, the initial oxidants, HO^•^ and/or L_x_(CO_3_)Fe^IV^=O, may oxidize the Fe-bound carbonate to ^•^CO_3_^−^ [69,70], which itself is a powerful yet more selective oxidant.

An important question is that of whether metal chelators can prevent the formation of oxidizing species during the reaction of peroxides with L_x_Fe^III^ and L_x_Fe^II^. Walling et al. [71] demonstrated the oxidation of a variety of organic substrates by (EDTA)Fe^III^/H_2_O_2_, providing evidence that oxidation reactions prevail in the presence of EDTA. Likewise, Graf et al. showed that EDTA did not prevent the oxidation of dimethylsulfoxide (DMSO) (ultimately to formaldehyde) induced by L_x_Fe^III^ and hypoxanthine/xanthine oxidase [72]. However, DTPA prevented the oxidation of DMSO, providing evidence that the chelator structure plays an important role in the efficiency of preventing substrate oxidation. These findings can, in part, be rationalized by the complex geometries of (EDTA)Fe^II^ and (EDTA)Fe^III^, where crystal structures demonstrate a distortion from octahedral geometry, resulting in the availability of a seventh binding site for a reaction to take place [73,74]. In aqueous solution, this seventh binding site generally coordinates with water [73,74], also indicated by a dissociable proton of (EDTA)Fe^III^(H_2_O) with pK_a_ ≈ 7.6 [75,76]. The bound water can be replaced by H_2_O_2_ [65,71,77] and, in case of (EDTA)Fe^II^, also by molecular oxygen [78]. In fact, (EDTA)Fe^II^ efficiently reacts with H_2_O_2_, with k > 3 × 10^3^ M^−1^s^−1^ [79,80].

### 3.3. Photochemical Generation of Radicals

Depending on the manufacturing environment and clinical use, protein formulations can be exposed to UVA and/or visible light [81], and an increasing number of studies show visible- or ambient-light-induced degradation of therapeutic proteins [58,82,83,84,85,86,87,88,89,90,91,92]. In particular, visible light photo-degradation is not easily rationalized with the known absorption characteristics of individual amino acids. This presents a challenge for the mechanistic analysis of processes leading to photo-degradation under visible light exposure, which is addressed in a recent review [93]. It is generally possible that photo-sensitizers are generated from the oxidative degradation of proteins and/or excipients, i.e., protein di-tyrosine from Tyr [94], 6a-hydroxy-2-oxo-octahydropyrollo[2,3-d]imidazole-5-carboxylic acid from His [17], advanced glycation end-products (AGEs) from the breakdown of carbohydrates [95], and cross-links between amino acids and lipid peroxidation products [96]. In addition, certain constituents or impurities present in cell culture media that co-purify with the protein may act as photo-sensitizers, e.g., riboflavin [97,98,99] or pterin derivatives [100,101,102]. These photo-sensitizers can generate radicals in pharmaceutical formulations via a type I process, which represents an electron transfer reaction by a photosensitizer, subsequent to which a radical intermediate reacts with oxygen [103] (in contrast, a type II process entails the generation of singlet oxygen, ^1^O_2_ [103]).

Tryptophan residues can form cation-π complexes [104,105,106], which absorb visible light [107,108]. In such complexes, the electron density is shared between the Trp π-system and the cation, resulting in spectroscopic properties reminiscent of Trp^•^ radicals [108]. Hence, Trp cation-π complexes may serve as chromophores suitable for the initiation of photo-degradation by visible light, a possibility that should be tested experimentally.

In view of the discussion of Fe^III^-dependent oxidation reactions in pharmaceutical formulations (see Section 3.2 above), the possibility of photo-Fenton reactions as a source of free radicals is a viable option. Pharmaceutical buffers (e.g., acetate, succinate, citrate) and amino acids contain carboxylate groups, where Fe^III^-carboxylate complexes are characterized by broad absorption bands in the UVA and visible regions. Under light exposure, these Fe^III^-carboxylate complexes can undergo ligand-to-metal-charge transfer (LMCT), reducing Fe^III^ to Fe^II^, and oxidizing the carboxylate ligand, which subsequently decarboxylates reactions (13) and (14) [109,110,111,112,113].
RCO_2_^−^-Fe^III^ → RCO_2_^•^-Fe^II^(13)
RCO_2_^•^-Fe^II^ → R^•^ + CO_2_ + Fe^II^(14)

The resulting carbon-centered radical R^•^ will add oxygen to yield a peroxyl radical, ROO^•^, unless R^•^ is ^•^CO_2_^−^ (see below), while Fe^II^ reduces O_2_ to O_2_^•−^ [65] and H_2_O_2_ [114]. With respect to the necessary concentrations of Fe^III^, basal levels of Fe^III^ in 10 mM citrate buffer, pH 6.0, were sufficient to promote the photo-oxidation of Met-enkephalin during near-UV photo-irradiation with a light dose of 25.2 Whm^−2^ [115], i.e., ca. 1/8 of the light dose required according the ICH Q1B guidelines for photostability studies [116]. In these experiments, various lots of citrate were tested, and the photo-oxidation yields from Met-enkephalin correlated with the basal Fe^III^ levels [115]. An important detail is the formation of ^•^CO_2_^−^ during the photo-irradiation of citrate-Fe^III^ with either near-UV or visible light, detected via spin-trapping with DMPO [115,117]. The ^•^CO_2_^−^ radical is a powerful reductant (E^o^(CO_2_/^•^CO_2_^−^) ≈ 1.93 ± 0.22 V vs. NHE [118]) that reduces Fe^III^ to Fe^II^ [119], O_2_ to O_2_^•−^ [119,120] and disulfide (RSSR) to a thiyl radical (RS^•^) and thiolate (RS^−^) [121,122,123].

Mechanistic studies suggest that the formation of ^•^CO_2_^−^ from citrate involves LMCT from the (deprotonated) citrate hydroxyl group rather than the citrate carboxyl groups, generating an intermediary alkoxyl radical (RO^•^), which undergoes α-β cleavage of the central carboxylate group (Figure 1; reactions 15 and 16) [117]. In reaction 15, the initial citrate-Fe^III^ complex is drawn with reference to the crystal structure of mononuclear (citrate)_2_Fe^III^ [124], which shows that the hydroxyl group is deprotonated.

A similar mechanism was recently observed for a monoclonal antibody (IgG1) in the presence of Fe^III^ and His buffer. In this case, photo-induced LMCT from a deprotonated Thr residue, Thr^259^, led to an intermediary Thr side chain alkoxyl radical (Figure 2, reaction 17), which underwent α-β cleavage, triggering side chain cleavage (Figure 2, reaction 18) and, ultimately, backbone fragmentation [92].

### 3.4. Generation of Radicals via Mechanical Stress

Protein formulations are exposed to various types of mechanical stress during manufacturing and transportation. Under certain circumstances, high shear stresses [125,126], mixing, pumping, filling [127,128,129] and mechanical shock [130,131,132] may lead to cavitation [133], a process that can cause the formation of HO^•^ radicals and even O atoms [134]. Hence, mechanical stresses have the potential to trigger the formation of highly oxidizing radicals, which can subsequently react with formulation constituents.

## 4. Protein Formulations Containing Additional Excipients

### 4.1. Formulations Containing Antimicrobial Preservatives

In order to ensure sterility, multidose formulations contain antimicrobial preservatives (APs) such as, e.g., phenol, m-cresol, benzyl alcohol, thimerosal or chlorobutanol [135,136] (for a summary of antimicrobial preservative-containing peptide and protein formulations listed in the *Physicians’ Desk Reference*, PDR, see [135]). Some of the common antimicrobial preservatives are susceptible to oxidative degradation, potentially generating radicals in pharmaceutical formulations.

The exposure of benzyl alcohol to air leads to the slow formation of benzaldehyde, Ph-CHO [136]. Benzaldehyde spontaneously oxidizes to benzoic acid [137]. The latter pathway involves the formation of an intermediary benzoylperoxyl radical, Ph-C(O)OO^•^ (reactions 19 and 20), where In^•^ represents an initiating radical [137]. However, the presence of benzyl alcohol can suppress benzoic acid formation via the reaction of the benzoylperoxyl radical with benzylalcohol to generate peroxybenzoic acid, Ph-C(O)OOH, and an α-hydroxybenzyl radical, Ph-C^•^H-OH (reaction 21) [137]. The reaction of the α-hydroxybenzyl radical with molecular oxygen will ultimately generate benzaldehyde and superoxide (reaction 22).
Ph-CHO + In^•^ → InH + Ph-^•^C(O)(19)
Ph-^•^C(O) + O_2_ → Ph-C(O)OO^•^(20)
Ph-C(O)OO^•^ + Ph-CH_2_OH → Ph-C(O)OOH + Ph-C^•^H-OH(21)
Ph-C^•^H-OH + O_2_ → Ph-CHO + H^+^/O_2_^•−^(22)

Therefore, formulations containing benzyl alcohol bear a potential risk for the formation of oxygen-centered radicals (peroxyl radicals, superoxide) and peroxides (peroxybenzoic acid).

The potential exposure of phenol and m-cresol to hydroxyl radicals (such as those generated by Fenton-type reactions; see Section 3.2 above) will lead to hydroxylation, preferentially in the ortho- or para-position with regard to the existing hydroxy substituent(s) [138,139,140]. Such hydroxylation reactions generate catechol derivatives, which can further promote Fenton-type reactions through redox cycling [141,142]. During redox cycling, a catechol derivative reduces L_x_Fe^III^ to L_x_Fe^II^, generating a semiquinone radical (Figure 3; equilibrium 23), which can further reduce O_2_ to O_2_^•−^ (equilibrium 24), generating a quinone derivative. The latter can comproportionate with a catechol to regenerate semiquinone derivatives (equilibrium 25) [142]. The dismutation of O_2_^•−^ will generate H_2_O_2_, which will regenerate L_x_Fe^III^ through a reaction with L_x_Fe^II^, generating HO^•^ radicals (see Section 3.2).

### 4.2. Formulations Containing Zn(II)

Specifically, insulin formulations, e.g., Humulin R^®^ or Humalog^®^, contain Zn(II), generally in the form of ZnO (see package inserts for Humulin N^®^ and Humalog^®^), which releases Zn^2+^ [143]. ZnO confers antimicrobial activity [143], but the released Zn^2+^ ions also support the formation of a native insulin hexamer [144]. Both Humulin N^®^ and Humalog^®^ also contain m-cresol and phenol, which are susceptible to hydroxylation and, subsequently, redox cycling (see Section 4.1). It was demonstrated that Zn^2+^ increased total phenol oxidation (monitored as total organic carbon, TOC) during the Fenton oxidation of phenols, which has been rationalized by a more persistent semiquinone radical as a result of Zn^2+^ complexation, generating more HO^•^ radicals [142]. Hence, the combination of phenols and Zn^2+^ may increase the susceptibility of a formulation to Fenton oxidation.

An alternative mechanism by which ZnO, specifically, may promote oxidation reactions is photo-degradation. ZnO is a semiconductor with a band gap of 3.2–3.7 eV [145], which would require light with wavelengths of λ = 387–335 nm to excite an electron from the valence band to the conduction band. Generally, the conduction band electron can reduce adsorbed O_2_ to O_2_^•−^, while the remaining positive hole, h^+^, in the valence band can oxidize adsorbed H_2_O/HO^−^ to the HO^•^ radical [145,146]. ZnO was tested as a photo-catalyst under light exposure with λ > 300 nm on a coated glass plate [147], showing greater activity than WO_3_, an activity comparable to that of brookite (TiO_2_), but an activity lower than that of anatase (TiO_2_). However, ZnO was more active than anatase in the photocatalytic degradation of humic acid in aqueous solution with pH 7.88 [148].

## 5. IV Enzyme Formulations for Enzyme Replacement Therapy

A review of IV formulations for enzyme replacement therapy [149,150] reveals that these formulations generally do not contain unusual excipients (for example, see package inserts for Aldurazyme^®^, Elaprase^®^, Vimizim^®^, Naglazyme^®^, Mepsevii^®^, VPRIV^TM^ or Nexviazyme^TM^). However, inspection of the active sites of the some of the relevant enzymes shows the presence of Cys residues, e.g., in N-acetylgalactosamine-6-sulfatase [151] and iduronate-2-sulfatase [152]. These Cys residues are post-translationally modified to C_α_-formylglycine (FGly) and, therefore, are not amenable to Cys oxidation.

## 6. Conclusions

Based on information on impurities and stress factors that affect pharmaceutical formulations, a series of mechanisms are formulated that could be responsible for free radical formation in pharmaceutical formulations. The focus of this article is on highly probable reactions; additional pathways may be possible in isolated cases when pharmaceutical formulations contain high levels of specific impurities that are not generally present. With respect to the design of stress tests for pharmaceutical formulations, highly probable reactions should be kept in mind. For example, it may be questionable whether the addition of Fe^II^ to a pharmaceutical formulation may generate information about the kinetics of iron-dependent oxidation degradation reactions under storage conditions, as iron impurities will likely be present as Fe^III^. However, the addition of Fe^II^ may lead to mechanistic information that can be used to predict certain degradation pathways in cases whereby Fe^III^ is converted to Fe^II^, for example, through reaction with H_2_O_2_ or hydroperoxides. A limitation of mechanistic investigations of radical-induced oxidation reactions in pharmaceutical formulations will always be that the precise quantity and nature of the radicals specifically generated under storage conditions are usually unknown. Even the monounsaturated oleic acid, the main component of polysorbate 80 fatty acid esters [35,36], can generate a number of different peroxyl radicals [40]. It is unknown to what extent the nature of these different peroxyl radicals would affect the kinetics of chain propagation within polysorbate 80 micelles. This question may be addressed through the quantification of specific reaction products that are representative of individual oxidation pathways, a task that may require the modification or improvement of analytical methodology, potentially supported by artificial intelligence.

## Data Availability

Not applicable.

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
