# Peer review of "Primary Processes of Free Radical Formation in Pharmaceutical Formulations of Therapeutic Proteins"

_biomolecules, 2023, doi:10.3390/biom13071142_

Round 1
Reviewer 1 Report
I enjoyed reading the review. My only suggestion is to include a paragraph or mini-subheading on iv enzyme formulations. Depending on the active site structure or prosthetic groups, there could be specific mechanisms of inactivation relevant to each zyme catalytic activity
Author Response
We have added a paragraph on IV enzyme formulations (Section 5). Inspection of the active sites of several of these enzymes reveal no specific liabilities; some contain Cys residues but these are post-translationally modified to Calpha-formylglycine, and, therefore, not amenable to Cys oxidation.
Reviewer 2 Report
The review is written in good scientific language. Known facts are clearly stated. The author himself notes that “The focus of this article is on highly probable reactions: additional mechanisms may be possible in isolated cases when pharmaceutical formulations contain high levels of specific impurities not generally present”. In this regard, the review is valuable.
However, unfortunately, not one specific example of the potential danger of free radical formation in pharmaceutical formulations of therapeutic proteins has been given. There are many drugs for which additives to acting recombinant proteins are known. For example, for the most important recombinant insulin protein (Insulin Lispro preparation), zinc oxide is included in the composition (as well as Metacresol, Glycerol,Disodium hydrogen phosphate heptahydrate, Water for injections, Hydrochloric acid (for pH adjustment), Sodium hydroxide (for pH adjustment)) https://www.medicines.org.uk/emc/product/1640/pil#gref . And zinc oxide can participate in the oxidation https://doi.org/10.3389/fenvs.2022.807290 , zinc dioxide can enter in modified Fenton reaction (https://doi.org/10.1007/s11164-022-04837-z). Perhaps, for each section, several specific examples of recombinant protein preparations used, where certain known additives are potentially dangerous from the point of view of free radical formation, should be given. Or it would be important to group a few representative cases with potentially harmful additives in terms of recombinant therapeutic protein oxidation into a table. This would greatly improve the overall meaning of the article.
Formulas of numerous equations are stretched along the line. Use the special form provided in the MDPI template to fill in the equations. This will make them much easier to understand.
The abstract is too short, only 62 words, and in my opinion does not fully disclose the information that the review is devoted to. Increase the size of the abstract, in particular, add a description of the terms that you included in the Keywords.
In general, the review has intrinsic value and originality; after the proposed corrections are made, it can be published.
Author Response
- We have included information on formulations containing specific additives. Rather than a table, we have added section 4, focusing on antimicrobial preservatives and their oxidation reactions in section 4.1, and on ZnO in section 4.2. There are many, especially multidose, formulations containing antimicrobial preservatives, and a list of marketed formulations is presented in reference [135] which we provide in the text. Attention is given in section 4.2. to the potential interplay of Zn(II) and phenolic antimicrobial preservatives, where Zn(II) may increase the susceptibility to oxidation. examples for such formulations are Humulin R and Humalog. We have also added section 5 on IV enzyme formulations.
- The equations have been modified
- We have expanded the abstract as requested by the reviewer.